# B-GAN: UNIFIED FRAMEWORK OF GENERATIVE ADVERSARIAL NETWORKS

**Masatosi Uehara, Issei Sato, Masahiro Suzuki, Kotaro Nakayama, Yutaka Matsuo**
The University of Tokyo
`uehara-masatoshi136@g.ecc.u-tokyo.ac.jp`
`sato@k.u-tokyo.ac.jp`
`{masa, nakayama, matsuo}@weblab.t.u-tokyo.ac.jp`

## ABSTRACT

Generative adversarial networks (GANs) are successful deep generative models. They are based on a two-player minimax game. However, the objective function derived in the original motivation is changed to obtain stronger gradients when learning the generator. We propose a novel algorithm that repeats density ratio estimation and f-divergence minimization. Our algorithm offers a new unified perspective toward understanding GANs and is able to make use of multiple viewpoints obtained from the density ratio estimation research, e.g. what divergence is stable and relative density ratio is useful.

## 1 INTRODUCTION

There have been many recent studies about deep generative models. Generative adversarial networks (GAN) (Goodfellow et al., 2014) is the variant of these models that has attracted the most attention. It has been demonstrated that generating vivid, realistic images from a uniform distribution is possible (Radford et al., 2015; Denton et al., 2015). GANs are formulated as a two-player minimax game. However, the objective function derived in the original motivation is modified to obtain stronger gradients when learning the generator. GANs have been applied in various studies; however, few studies have attempted to reveal their mechanism (Goodfellow, 2014; Huszar, 2015).

Recently, f-GAN, which minimizes the variational estimate of f-divergence, has been proposed (Nowozin et al., 2016). The original GAN is a special case of f-GAN.

In this study, we propose a novel algorithm inspired by GANs from the perspective of density ratio estimation based on the Bregman divergence, which we refer to as b-GAN. The proposed algorithm iterates density ratio estimation and f-divergence minimization based on the obtained density ratio. This study make the following two primary contributions:

1. We derive a novel unified algorithm that employs well-studied results regarding density ratio estimation (Kanamori et al., 2012; Sugiyama et al., 2012; Menon & Ong, 2016).

2. In the original GANs, the value function derived from the two-player minimax game does not match the objective function that is actually used for learning the generative model. In our algorithm, the objective function derived from the original motivation is not changed for learning the generative model.

Table 1: Relation among GAN, f-GAN and b-GAN.

| Name | D-step (updating $\theta_D$) | G-step (updating $\theta_G$) |
|---|---|---|
| GAN | Estimate $\frac{p}{p+q}$ | Adversarial update |
| f-GAN | Estimate $f'(\frac{p}{q})$ | Minimize a part of variational |
| | when $f = x \log x - (x+1)\log(x+1)$, it is a GAN | Estimate of f-divergence |
| b-GAN | Estimate $\frac{p}{q} = r(x)$ | $\min_\theta E_{x \sim q(x;\theta)}[f(r(x))]$ |
| (this work) | Dual relation with f-GAN | Minimize f-divergence directly |

The remainder of this study is organized as follows. Section 2 describes related work. Section 3 introduces and analyzes the proposed algorithm in detail. Section 4 explains the proposed algorithm for specific cases. Section 5 reports experimental results. Section 6 summarizes our findings and discusses future work.

## 2 RELATED WORK

In this study, we denote an input space as $X$ and a hidden space as $Z$. Let $p(x)$ be the distribution of training data over $X$ and $q(x)$ be the generated distribution over $X$.

GANs (Goodfellow et al., 2014) were developed based on a game theory scenario, where two model, i.e., a generator network and a discriminator network, are simultaneously trained. The generator network $G_{\theta_G}(z)$ produces samples with a probability density function of $q(x; \theta_G)$. The discriminator network $T_{\theta_D}(x)$ attempts to distinguish the samples from the training samples and that from the generator. GANs are described as a zero-sum game, where the function $v(G, T)$ determines the pay-off of the discriminator and the function $-v(G, T)$ determines the pay-off of the generator. The discriminator $T_{\theta_D}(x)$ and generator $G_{\theta_G}(z)$ play the following two-player minimax game $\min_{\theta_G} \max_{\theta_D} v(G, T)$, where $v(G, T)$ can be expressed as follows:

$$E_{x \sim p(x)}[\log T_{\theta_D}(x)] + E_{x \sim q(x;\theta_G)}[\log(1 - T_{\theta_D}(x))].$$

The discriminator and generator are iteratively trained by turns. For fixed G, the optimal T(x) is $\frac{p(x)}{p(x)+q(x)}$. This suggests that training the discriminator can be formulated as a density ratio estimation. The generator is trained to minimize $v(G, T)$ adversarially. In fact, maximizing $E_{x \sim q(x;\theta_G)}[\log T_{\theta_D}(x)]$ is preferable instead of minimizing $E_{x \sim q(x;\theta_G)}[\log(1 - T_{\theta_D}(x))]$. Although this does not match the theoretical motivation, this heuristic is the key to successful learning. We analyze this heuristic in Section 3.4.

f-GAN (Nowozin et al., 2016) generalizes the GAN concept. First, we introduce f-divergence (Ali & Silvey, 1966). The f-divergence measures the difference between two probability distributions $p$ and $q$ and is defined as

$$D_f(p||q) = \int q(x) f\left(\frac{p(x)}{q(x)}\right) dx = \int q(x) f(r(x)) \, dx, \tag{1}$$

where $f(x)$ is a convex function satisfying $f(1) = 0$. Note that in the space of positive measures, i.e., not satisfying normalized conditions, f-divergence must satisfy $f'(1) = 0$ due to its invariance (Amari & Cichoki, 2010).

The function $v(G, T)$ of f-GAN is given by

$$E_{x \sim p(x)}[T_{\theta_D}(x)] - E_{x \sim q(x;\theta_G)}[f^*(T_{\theta_D}(x))], \tag{2}$$

where $f^*$ is a Fenchel conjugate of $f$ (Nguyen et al., 2010). In Eq. 2, $v(G, T)$ comes from,

$$\int q(x) f\left(\frac{p(x)}{q(x)}\right) dx = \sup_T E_{x \sim p}[T(x)] - E_{x \sim q}[f^*(T(x))]. \tag{3}$$

Following GANs, $\theta_D$ is trained to maximize Eq. 2 in order to estimate the f-divergence. In contrast, $\theta_G$ is trained to adversarially minimize Eq. 2 to minimize the f-divergence estimate. However, as in GANs, maximizing $E_{x \sim q}[T(x)]$ is used rather than minimizing $E_{x \sim q}[-f^*(T(x))]$. The latter optimization is theoretically valid in their formulation; however, they used the former heuristically. Similar to GANs, f-GAN also formulates the training discriminator as a density ratio estimation. For a fixed G, the optimal $T(x)$ is $f'(\frac{p}{q})$, where $f'$ denotes the first-order derivative of $f$. When $f(x)$ is

$x \log x - (x+1) \log(1+x)$, f-GANs are equivalent to GANs. Table 1 summarizes GAN and f-GAN. We denote the step for updating $\theta_D$ as D-step and the step for updating $\theta_G$ as G-step.

## 3 METHOD

As described in Section 2, training the discriminators in the D-step of GANs and f-GANs is regarded as density ratio estimation. In this section, we further extend this idea. We first review the density ratio estimation method based on the Bregman divergence. Then, we explain and analyze a novel proposed b-GAN algorithm. See appendix E for recent research related to density ratio estimation.

### 3.1 DENSITY RATIO MATCHING UNDER THE BREGMAN DIVERGENCE

There have been many studies on direct density ratio estimation, where a density ratio model is fitted to a true density ratio model under the Bregman divergence (Sugiyama et al., 2012). We briefly review this method.

Assume there are two distributions $p(x)$ and $q(x)$. Our aim is to directly estimate the true density ratio $r(x) = \frac{p(x)}{q(x)}$ without estimating $p(x)$ and $q(x)$ independently . Let $r_\theta(x)$ be a density ratio model. The integration of the Bregman divergence $\mathfrak{B}_f[r(x)\|r_\theta(x)]$ between the density ratio model and the true density ratio with respect to measure $q(x)dx$ is

$$
\begin{aligned}
BD_f(r\|r_\theta) &= \int \mathfrak{B}_f[r(x)\|r_\theta(x)]q(x)dx \\
&= \int \left( f(r(x)) - f\left(r_\theta(x)\right) - f'\left(r_\theta(x)\right)\left(r(x) - r_\theta(x)\right) \right) q(x)dx.
\end{aligned}
\tag{4}
$$

We define the terms related to $r_\theta$ in $BD_f(r\|r_\theta)$ as

$$
\begin{aligned}
BR_f(r_\theta) &= \int \left( f'(r_\theta(x))r_\theta(x) - f(r_\theta(x)) \right) q(x)dx - \int f'\left(r_\theta(x)\right) p(x)dx \tag{5} \\
&= \int f'(r_\theta(x)) \left( r_\theta(x)q(x) - p(x) \right) dx - D_f(qr_\theta\|q). \tag{6}
\end{aligned}
$$

Thus, estimating the density ratio problem turns out to be the minimization of Eq. 5 with respect to $\theta$.

### 3.2 MOTIVATION

In this section, we introduce important propositions required to derive b-GAN. Proofs of propositions are given in Appendix C. The following proposition suggests that the supremum of the negative of Eq. 5 is equal to the f-divergence between $p(x)$ and $q(x)$.

**Prop 3.1.** *The following equation holds:*

$$
E_q\left[ f\left( \frac{p(x)}{q(x)} \right) \right] = \sup_{r_\theta} E_{x\sim p}[f'\left(r_\theta(x)\right)] - E_{x\sim q}[(f'(r_\theta(x))r_\theta(x) - f(r_\theta(x)))].
\tag{7}
$$

*The right side of Eq. 7 reaches the supremum when $r_\theta(x) = r(x)$ is satisfied.*

It has been shown that the supremum of negative of Eq. 5 is equivalent to the supremum of Eq. 2. Interestingly, the negative of Eq. 5 has a dual relation with the objective function of f-GAN, i.e., Eq. 2.

**Prop 3.2.** *Introducing dual coordinates $T_{\theta_D} = f'(r_\theta)$(Amari & Cichoki, 2010) yields the right side of Eq. 5 from Eq. 2.*

Prop 3.2 shows that the D-step of f-GAN can be regarded as the density ratio estimation because Eq. 5 expresses the density ratio estimation and Eq. 2 is a value function of f-GAN.

### 3.3 B-GAN

Our objective is to minimize the f-divergence between the distribution of the training data $p(x)$ and the generated distribution $q(x)$. We introduce two functions constructed using neural networks: $r_{\theta_D}(x) : X \to \mathcal{R}$ parameterized by $\theta_D$, and $G_{\theta_G}(z) : Z \to X$ parameterized by $\theta_G$. Measure $q(x; \theta_G)dx$ is a probability measure induced from the uniform distribution by $G_{\theta_G}(z)$. In this case, $r_{\theta_D}(x)$ is regarded as a density ratio estimation network and $G_{\theta_G}(z)$ is regarded as a generator network for minimizing the f-divergence between $p(x)$ and $q(x)$.

Motivated by Section 3.2, we construct a b-GAN using the following two steps.

1. Update $\theta_D$ to estimate the density ratio between $p(x)$ and $q(x; \theta_G)$. To achieve this, we minimize Eq. 5 with respect to $r_\theta(x)$. In this step, the density ratio model $r_\theta(x)$ in Eq. 5 can be considered as $r_{\theta_D}(x)$ in this step.

2. Update $\theta_G$ to minimize the f-divergence $D_f(p||q)$ between $p(x)$ and $q(x; \theta_G)$ using the obtained density-ratio. We are able to suppose that $q(x; \theta_G)r_\theta(x)$ is close to $p(x)$. Instead of $D_f(p||q)$, we update $\theta_G$ to minimize the empirical approximation of $D_f(qr_\theta||q)$.

The b-GAN algorithm is summarized in Algorithm 1, where B is the batch size. In this study, a single-step gradient method (Goodfellow et al., 2014; Nowozin et al., 2016) is adopted.

---

**Algorithm 1:** b-GAN

**for** number of training iterations **do**

sample $\hat{X} = \{x_1, ..., x_B\}$ from $p(x)$ and $\hat{Z} = \{z_1, ..., z_B\}$ from an uniform distribution.

D-step: Update $\theta_D$:

$$\theta_D^{t+1} = \theta_D^t - \nabla_{\theta_D} \left( \frac{1}{B} \sum_{i=1}^{B} f'\big(r_{\theta_D}(G(z_i))\big)r_{\theta_D}(G(z_i)) - f\big(r_{\theta_D}(G(z_i))\big) - f'\big(r_{\theta_D}(x_i)\big) \right).$$

G-step: Update $\theta_G$:

$$\theta_G^{t+1} = \theta_G^t - \nabla_{\theta_G} \left( \frac{1}{B} \sum_{i=1}^{B} f\big(r(G_{\theta_G}(z_i))\big) \right).$$

**end for**

---

In the D-step, the proposed algorithm estimates $\frac{p}{q}$ toward any divergence; thus, it differs slightly from the D-step of f-GAN because the estimated values, i.e., $f'(\frac{p}{q})$, are dependent on the divergences. We also introduce an f-GAN-like update as follows. As mentioned in Section 2, we have two options in the G step.

1. D-step: minimize $E_{x \sim p(x)}[-f'(r_{\theta_D}(x))] + E_{x \sim q(x)}[f'(r_{\theta_D}(x))r_{\theta_D}(x) - f(r_{\theta_D}(x))]$ w.r.t $\theta_D$.

2. G-step: minimize $E_{x \sim q(x; \theta_G)}[-f'(r(x))]$ or $E_{x \sim q(x; \theta_G)}[-f'(r(x))r(x) + f(r(x))]$ w.r.t $\theta_G$.

### 3.4 ANALYSIS

Following Goodfellow et al. [2014], we explain the validity of the G-step and D-step. We then explain the meaning of b-GAN. Finally, we analyze differences between b-GAN and f-GAN.

The density ratio is estimated in the D-step. The estimator of $r(x)$ is an M-estimator and is asymptotically consistent under the proper normal conditions (Appendix E.1).

In the G-step, we update the generator as minimizing $D_f(p||q)$ by replacing $p(x)$ with $r_\theta(x)q(x)$. We assume that $q(x; \theta_G)$ is equivalent to $p(x)$ when $\theta_G = \theta^*$, $q(x; \theta_G)$ is identifiable, and the optimal $r(x)$ is obtained in the D-step. By our assumption, the acquired value in the G-step is $\hat{\theta}$, which minimizes the empirical approximation of $D_f(r(x)q(x; \theta_G)||q(x; \theta_G)) = E_{x \sim q(x; \theta_G)}[r(x)]$.

When $\theta_G$ is equal to $\theta^*$, this equation is equal to 0. The estimator $\hat{\theta}$ can be considered as a kind of Z-estimator.

Usually, we cannot perform only a G-step because we do not know the form of $p(x)$ and $q(x)$. In b-GAN, $D_f(p\|q)$ can be minimized by estimating the density ratio $r(x)$ without estimating the densities directly.

In fact, the $r(x)$ obtained at each iteration is different and not optimal because we adopt a single-step gradient method (Nowozin et al., 2016). Thus, b-GAN dynamically updates the generator to minimize the f-divergence between $p(x)$ and $q(x)$. As mentioned previously, $f(x)$ must satisfy $f'(1) = 0$ in this case because we cannot guarantee that $r_\theta(x)q(x)$ is normalized.

Similar to GANs, the D-step and G-step work adversarially. In the D-step, $r_\theta(x)$ is updated to fit the ratio between $p(x)$ and $q(x)$. In the G-step, $q(x)$ changes, which means $r_\theta(x)$ becomes inaccurate in terms of the density ratio estimator. Next, $r_\theta(x)$ is updated in the D-step so that it fits the density ratio of $p(x)$ and the new $q(x)$. This learning situation is derived from Eq. 6 , which shows that $\theta_D$ is updated to increase $D_f(qr_\theta\|q)$ in the D-step. In contrast, $\theta_G$ is updated to decrease $D_f(qr_\theta\|q)$ in the G-step.

In Section 3.3, we also introduced a f-GAN-like update. Three choices can be considered for the G-step:

$$(1)E_{x\sim q(x;\theta_G)}[f(r(x))], (2)E_{x\sim q(x;\theta_G)}[-f'(r(x))], (3)E_{x\sim q(x;\theta_G)}[-f'(r(x))r(x) + f(r(x))].$$

Note that $f$ is a convex function, $f(1) = 0$, and $f'(1) = 0$. It is noted in (Nowozin et al., 2016) that case (2) works better than case (3) in practice. We also confirm this. The complete reason for this is unclear. However, we can find a partial reason by differentiating objective functions with respect to $r$. The derivatives of the objective functions are

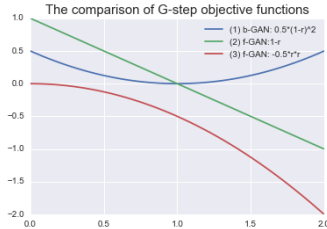

$$(1)f'(r), (2) - f''(r), (3) - rf''(r).$$

All signs are negative when $r(x)$ is less than 1. Usually, when $x$ is sampled from $q(x)$, $r(x)$ is less than 1. Therefore, we speculate that $r(x)$ is less than 1 during most of the learning process when $x$ is sampled from

Figure 1: The graph of (1), (2), (3) when $f$ is Pearson divergence

$q(x)$. When $r(x)$ is small, the derivative is also small in (3) because the term $r(x)$ is multiplied. Therefore, the derivative tends to be small in (3). The mechanism pulling $r(x)$ to 1 does not work when $r(x)$ is small. Thus, the case of (3) does not work well. A similar argument was proposed by Goodfellow et al. (2014) and Nowozin et al. (2016).

In our experimental case of (1) and (2) work properly (Section 5). The reason case (2) works is that function $-f'(r)$ behaves like an f-divergence and the derivative is large when $r(x)$ is small. However, we cannot guarantee that $-f'(r)$ satisfies the conditions of f-divergence between positive measures, i.e, $-f'(r)$ is a convex function and $-f''(1) = 0$. If the derivatives in case (2) are negative when $r(x)$ is greater than 1, there is a possibility that the mechanism pulling $r(x)$ to 1 does not occur. In contrast, in case (1), when $r(x)$ is greater than 1, the derivatives are positive, therefore, the mechanism pulling $r(x)$ to 1 occurs. This prevents generators from emitting the same points. We can expect the same effects as -minibatch discrimination- (Salimans et al., 2016).

Throughout the analysis, we can easily extend the algorithm of b-GAN by using different divergences in the G-step and D-step. The original GAN can be regarded as one of such algorithms.

## 4   ALGORITHMS FOR SPECIFIC CASES

We adopt $\alpha$-divergence in the positive measure space as the f-divergence (Amari & Cichoki, 2010). Here, we explain specific algorithms when $f$ is $\alpha$-divergence. Then, we explain some heuristics used in b-GAN.

### 4.1 Algorithms with $\alpha$-divergence

In $\alpha$-divergence, $f(r)$ in Eq. 1 is

$$
f_\alpha(r) = \begin{cases} \frac{4}{1-\alpha^2}(1 - r^{\frac{1+\alpha}{2}}) + \frac{2}{1-\alpha}(r-1) & (\alpha \neq \pm 1) \\ r \log r - r + 1 & (\alpha = 1) \\ -\log r + r - 1 & (\alpha = -1). \end{cases} \tag{8}
$$

The objective function derived from $\alpha$-divergence is summarized as follows.

Table 2: The summary of objective function

| alpha | D-step | G-step |
|---|---|---|
| 1 (KL divergence) | $E_{x \sim q(x)}[r_{\theta_D}(x) - 1] - E_{x \sim p(x)}[\log r_{\theta_D}(x)]$ | $E_{x \sim q(x;\theta_G)}[r(x) \log r(x) - r(x) + 1]$ |
| 3 (Pearson divergence) | $E_{x \sim q(x)}[0.5 r_{\theta_D}(x)^2 - 0.5] - E_{x \sim p(x)}[r_{\theta_D}(x) - 1]$ | $E_{x \sim q(x;\theta_G)}[0.5(r(x) - 1)^2]$ |
| -1 (Reversed KL divergence) | $E_{x \sim q(x)}[\log r_{\theta_D}(x)] - E_{x \sim p(x)}[-\frac{1}{r_{\theta_D}(x)}]$ | $E_{x \sim q(x;\theta_G)}[-\log(r(x)) + r(x) - 1]$ |

- $\alpha = -1$ In this case, $\alpha$-divergence is a Kullback-Leibler (KL) divergence. Density ratio estimation via the KL divergence corresponds to the Kullback-Leibler Importance Estimation Procedure (Sugiyama et al., 2012). In the G-step of an f-GAN-like update, the objective function is $E_{x \sim q(x;\theta_G)}[-\log r(x)]$ or $E_{x \sim q(x;\theta_G)}[1 - r(x)]$.

- $\alpha = 3$ Density ratio estimation via the Pearson divergence corresponds to the Least-Squares Importance Fitting (Yamada et al., 2011). It is more robust than under KL divergence (Yamada et al., 2011; Dawid et al., 2015). This is because Pearson divergence does not include the $\log$ term. Hence the algorithm using Pearson divergence should be more stable. In the G-step of f-GAN-like update, the objective function is $E_{x \sim q(x;\theta_G)}[1 - r(x)]$ or $E_{x \sim q(x;\theta_G)}[0.5 - 0.5 r(x)^2]$.

- $\alpha = -1$ Estimating the density ratio using reversed KL divergence seems to be unstable because reversed KL-divergence is mode seeking and the generated distribution changes at each iteration. However, it is preferable to use reversed KL divergence when generating realistic images (Huszar, 2015). In the G-step of an f-GAN-like update, the objective function is $E_{q(x;\theta_G)}[\frac{1}{r(x)}]$ or $E_{q(x;\theta_G)}[-\log r(x)]$.

### 4.2 Heuristics

We describe some heuristic methods that work for our experiments. The heuristics introduced here are justified theoretically in Appendix C.

In the initial learning process, empirical distribution $p$ and generated distribution $q$ are completely different. Therefore, the estimated density ratio $r(x) = \frac{p(x)}{q(x)}$ is enormous when $x$ is taken from $p$ and tiny when $x$ is taken from $q$. It seems that the learning does not succeed in this case. In fact, in our setting, when the final activation function of $r_{\theta_D}(x)$ is taken from functions in the range $(0, \infty)$, b-GAN does not properly work. Therefore, we use a scaled sigmoid function such as a two-times sigmoid function. A similar idea has also been used in (Cortes et al., 2010).

As mentioned, density ratio $\frac{p(x)}{q(x)}$ is extremely sensitive. To avoid this problem, in the D-step of the KL-divergence, we also conducted experiments wherein we estimated $\frac{p}{\alpha p + (1-\alpha)q}$ (where $\alpha$ is small.) rather than $\frac{p(x)}{q(x)}$. The same idea is introduced in the covariate shift situation (Sugiyama et al., 2013). A similar idea has also been used for GAN learning (Salimans et al., 2016) and class probability estimation (Reid & Williamson, 2010).

## 5 EXPERIMENTS

We conducted experiments to establish that the proposed algorithm works properly and can successfully generate natural images. The proposed algorithm is based on density ratio estimation; therefore, knowledge regarding the density ratio estimation can be utilized. In the experiments, using the Pearson divergence and estimating the relative density ratio is shown to be useful for stable learning. We also empirically confirm our statement in Section 3.4, i.e., f-divergence is increased when learning $\theta_D$ and decreased when learning $\theta_G$.

### 5.1 SETTINGS

We applied the proposed algorithm to the CIFAR-10 data set (Krizhevsky, 2009) and Celeb A data set (Liu et al., 2015) because they are often used in GAN research (Salimans et al., 2016; Goodfellow et al., 2014). The images size are $32 \times 32$ pixels. All results in this section are analyzed based on the results of the CIFAR-10 data set. The results for the Celeb A data set are presented in Appendix B. Our network architecture is nearly equivalent to that of previous study (Radford et al., 2015) (refer to the appendix A,B for details). Note that unless stated otherwise, the last layer function of $r_{\theta_D}(x)$ is a sigmoid function multiplied by two. We used the TensorFlow for automatic differentiation (Abadi et al., 2015). For stochastic optimization, Adam was adopted (Kingma & Ba, 2014).

### 5.2 RESULTS

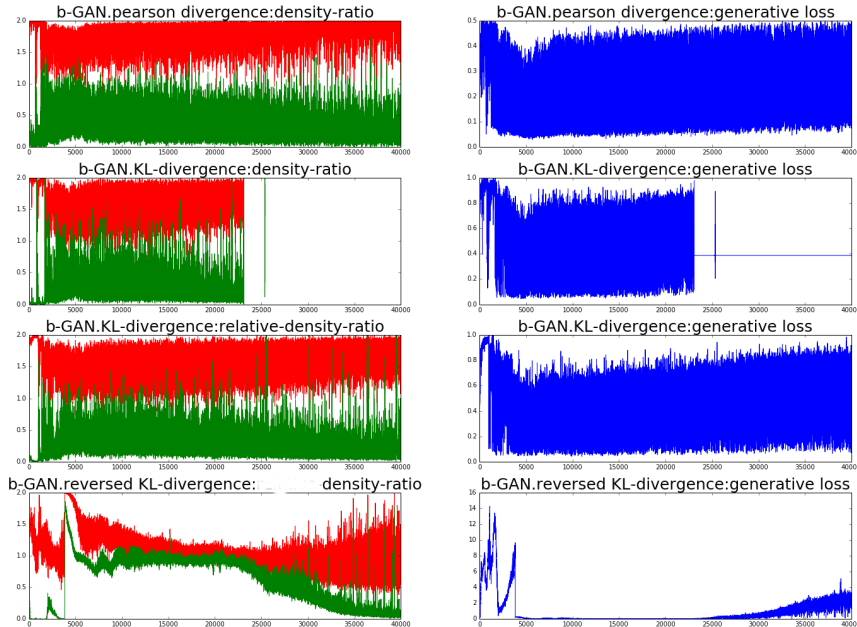

Figure 2: Comparative results: estimated density ratio values $r_{\theta_D}(x)$ from the training data (red), the estimated density-ratio values $r_{\theta_D}(x)$ from the generated distribution (green), generator losses taken in the D-step and G-step (blue). The top, second, and bottom rows show $r_{\theta_D}(x)$ and the losses of b-GAN with the Pearson divergence, KL divergence, modified KL divergence (relative density ratio estimation version, $\alpha = 0.2$), and reversed KL divergence, respectively.

Figure 2 shows the density ratio estimate $r_{\theta_D}(x)$ and loss values of the generators. For each divergence, we conducted four experiments with 40,000 epochs, where the initial learning rate value was fixed ($5 \times 10^{-5}$) with the exception of reversed KL divergence. These results show that the b-GANs using Pearson divergence are stable because the learning did not stop. The same results have been reported in the research into density ratio estimation (Yamada et al., 2011). In contrast, b-GANs using the KL divergence are unstable. In fact, the learning stopped between the 20,000th and 37,000th epoch when the learning rate was not as small. When we use a heuristic method, i.e.,

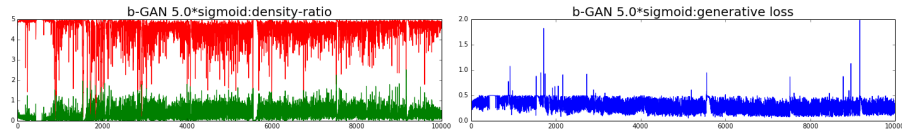

Figure 3: Density ratio value $r_{\theta_D}(x)$ and generator losses of b-GAN when the last output function is a sigmoid function multiplied by 5.

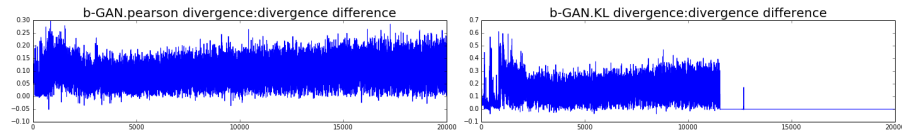

Figure 4: Divergence differences between D-step and G-step: b-GAN with Pearson divergence (left), b-GAN with KL divergence (right) .

estimating the relative density ratio as described in Section 4.2, this problem is solved. For reversed KL divergence, the learning stopped too soon if the initial learning rate value was $5 \times 10^{-5}$. If the learning rate was $1 \times 10^{-6}$, the learning did not stop; however, it was still unstable.

In Figure 2, the last layer activation function of the b-GANs is a twofold sigmoid function. In Figure 3, we use a sigmoid function multiplied by five. The results indicate that the estimated density ratio values approach one. We also confirm that the proposed algorithm works with sigmoid functions at other scales.

Figure 4 shows the estimated f-divergence $D_f(qr_\theta||q)$ before the G-step subtracted by $D_f(qr_\theta||q)$ after the G-step. Most of the values are greater than zero, which suggests f-divergence decreases at every G-step iteration. This observation is consistent with our analysis in Section 3.4.

Note that learning is successful with an f-GAN-like update when minimizing $E_q[-f'(r)]$. However, the learning f-GAN-like update when minimizing $E_q[f(r) - rf'(r)]$ did not work well for our network architecture and data set.

## 6    CONCLUSIONS AND FUTURE WORK

We have proposed a novel unified algorithm to learn a deep generative model from a density ratio estimation perspective. Our algorithm provides the experimental insights that Pearson divergence and estimating relative density ratio are useful to improve the stability of GAN learning. Other insights regarding density ratio estimation would also be also useful. GANs are sensitive to data sets, the form of the network and hyper-parameters. Therefore, providing methods to improve GAN learning is meaningful.

Related research to our study that focuses on linking density ratio and a GAN, has been performed by explaining specific algorithms independently (Mohamed & Lakshminarayanan, 2016). In contrast, our framework is more unified.

In future, the following things should be considered.

- What is the optimal divergence? In research regarding density ratio estimation, the Pearson divergence ($\alpha = 3$) is considered robust (Nam & Sugiyama, 2015). We empirically and theoretically confirmed the same property when learning deep generative models. It is also reported that using KL-divergence and reversed KL-divergence is not robust as scoring rules (Dawid et al., 2015). For generating realistic images, the reversed KL divergence ($\alpha = -1$) is preferred because it is mode seeking (Huszar, 2015). However, if $\alpha$ is small, the density ratio estimation becomes inaccurate. For a robust density ratio, using power divergence has also been proposed (Sugiyama et al., 2012). The determination of the optimal divergence is a persistent problem (Appendix E).

- What should be estimates in D-step? In the D-step of b-GAN, $r(x)$ is estimated. However, in the original GAN, $r(x)/(1 + r(x))$ is estimated. As unnormalized models, the latter is more robust than estimating $r(x)$ (Pihlaja et al., 2010) (Appendix E.7).

- We can consider algorithms that use different divergences in the G-step and D-step. In that case, choice of the divergences are more diverse. Original GANs are described in such algorithms as mentioned in Section 3.5.

- We can consider algorithms that use multiple divergences. This may improve the stability of learning.

- When sampling from $q(x)$, if the objective is sampling from real data $p(x)$, $r(x)$ should be multiplied. Hence, the density ratio is also useful when using samples from $q(x)$. How to use samples obtained from generators meaningfully is an remaining important problem.

ACKNOWLEDGEMENTS

The authors would like to thank Masanori Misono for technical assistance with the experiments. We are grateful to Masashi Sugiyama, Makoto Yamada, and the members of the Preferred Networks team.

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

# A    CIFAR-10 DATASET

Figure 5 shows samples generated randomly using b-GANs. These results indicate that b-GANs can create natural images successfully. We did not conduct a Parzen window density estimation for the evaluations because of Theis et al., [2016].

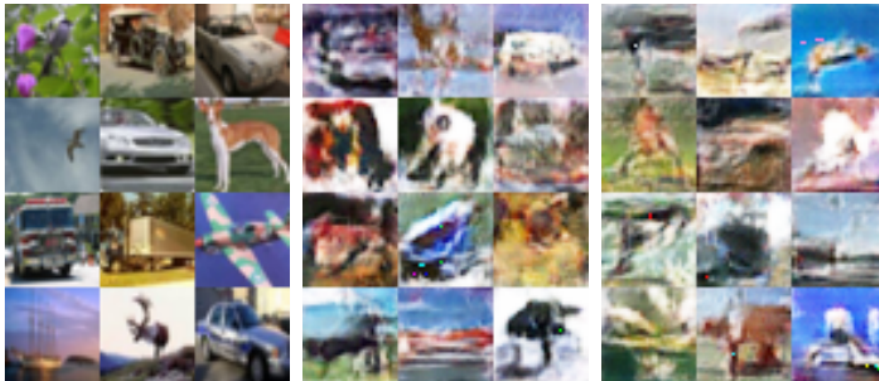

Figure 5: (Left) original images set, (middle) a set of images generated based on Pearson divergence, and (right) a set of images based on the KL divergence.

Here, we describe the network architecture of $r_{\theta_D}(x)$ and $G_{\theta_G}(z)$ used in the b-GAN. BN is the batch normalization layer (Sergey & Christian, 2015).

## A.1    $r_{\theta_D}(x)$

x $\to$ Conv(3, 64) $\to$ lRelu $\to$ Conv(64, 256) $\to$ BN $\to$ lRelu $\to$ Conv(256, 512) $\to$ BN $\to$ lRelu $\to$ Reshape($4 \times 4 \times 512$) $\to$ Linear($4 \times 4 \times 512$, 1) $\to$ $2 \times Sigmoid$

## A.2    $G_{\theta_G}(z)$

z $\to$ Linear(100,$4 \times 4 \times 512$) $\to$ BN $\to$ Relu $\to$ Reshape(4, 4, 512) $\to$ Conv(512, 256) $\to$ BN $\to$ Relu $\to$ Conv(256, 64) $\to$ BN $\to$ Relu $\to$ Conv(64, 3) $\to$ tanh

# B    CELEB A DATASET

We also applied our algorithm to the Celeb A data set. The images are resized and cropped to $64 \times 64$ pixcels. Figures 6 and 7 show samples randomly generated using b-GANs. The Network architecture is as follows.

## B.1    $r_{\theta_D}(x)$

x $\to$ Conv(3, 64) $\to$ lRelu $\to$ Conv(64, 128) $\to$ BN $\to$ lRelu $\to$ Conv(128, 256) $\to$ BN $\to$ lRelu $\to$ Conv(256, 512) $\to$ BN $\to$ lRelu $\to$ Reshape($4 \times 4 \times 512$) $\to$ Linear($4 \times 4 \times 512$, 1) $\to$ $2 \times$Sigmoid

## B.2    $G_{\theta_G}(z)$

z $\to$ Linear(64, $4 \times 4 \times 512$) $\to$ BN $\to$ Relu $\to$ Reshape(4, 4, 512) $\to$ Conv(512, 256) $\to$ BN $\to$ Relu $\to$ Conv(256, 128) $\to$ BN $\to$ Relu $\to$ Conv(128, 64) $\to$ BN $\to$ Relu $\to$ Conv(64, 3) $\to$ tanh

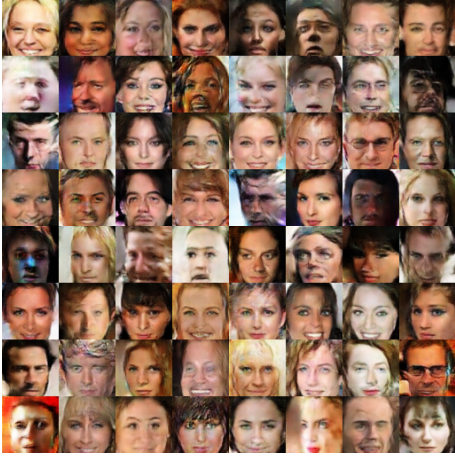
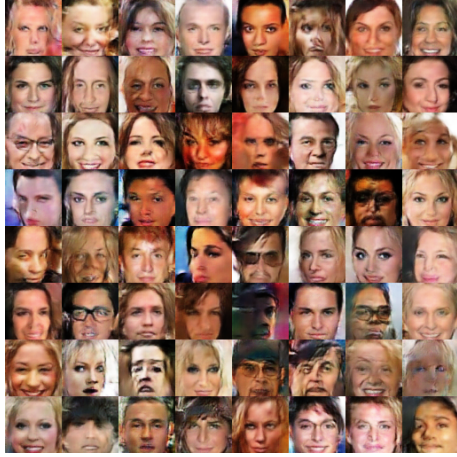

Figure 6: Pearson divergence                    Figure 7: KL divergence

## C    PROOF OF PROPOSITIONS IN SECTION 3

### C.1    PROOF OF PROP 3.1

From Eq. 4, the following equation holds,

$$E_{x \sim p}\left[f'\left(r_\theta(x)\right)\right] - E_{x \sim q}[(f'(r_\theta(x))r_\theta(x) - f(r_\theta(x)))] = -BD_f(r||r_\theta) + E_q\left[f\left(\frac{p(x)}{q(x)}\right)\right]. \quad (9)$$

Using $BD_f(r||r_\theta) \geq 0$ yields Eq. 7. We have $BD_f(r||r_\theta) = 0$ when $r$ is equal to $r_\theta$. Thus, the equality holds if and only if $r$ is equal to $r_\theta$.

### C.2    PROOF OF PROP3.2

$$
\begin{aligned}
\text{r.h.s of Eq.2} &= E_{x \sim p(x)}[f'(r_\theta(x))] - E_{x \sim q(x)}[f^*\left(f'(r_\theta(x))\right)] && (10) \\
&= E_{x \sim p(x)}[f'(r_\theta(x))] - E_{x \sim q(x)}[f(r_\theta(x))r_\theta(x) - f(r_\theta(x))] \\
&= E_{x \sim p(x)}[f'\left(r_\theta(x)\right)] - E_{x \sim q(x)}[(f'(r_\theta(x))r_\theta(x) - f(r_\theta(x)))] \\
&= \text{the negative of Eq. 5}
\end{aligned}
$$

In the derivation, we have used the following equation $f^*\left(f'(r_\theta(x))\right) = f(r_\theta(x))r_\theta(x) - f(r_\theta(x))$ .

## D    EXPLANATION OF HEURISTICS USING RADEMACHER COMPLEXITY

Here, we justify three points using Rademacher complexity. All techniques used are described in the literature (Mohri et al., 2012).

- Pearson divergence is preferable to KL divergence in density ratio estimation.
- Relative density ratio is useful (introduced in Sec 4.2).
- The meaning of bounding the last output functions of discriminators (introduced in Sec 4.2).

Our objective is estimating $r(x) = \frac{p(x)}{q(x)}$. We define hypothesis sets as $\mathbb{H}$. In Section 3.4, we assume $\mathbb{H}$ as parametric models for simplicity. In this section, we do not restrict $\mathbb{H}$ to parametric models. We define $r_0$ and $r_s$ as

$$r_0 = argmin_{h \in \mathbb{H}} BR_f(h)$$
$$r_s = argmin_{h \in \mathbb{H}} \hat{BR}_f(h),$$

where $\hat{BR}$ is an empirical approximation of $BR$. Note that $BR_f(h)$ reaches minimum values if and only if $r = h$ holds. We want to analyze $BR_f(r_s) - BR_f(r)$. The regret, i.e., $BR_f(r_s) - BR_f(r)$ can be bounded as follows:

$$BR_f(r_s) - BR_f(r) = BR_f(r_s) - BR_f(r_0) + BR_f(r_0) - BR_f(r)$$
$$\leq 2 \sup_{h \in \mathbb{H}} |BR_f(h) - \hat{BR}_f(h)| + BR_f(r_0) - BR_f(r). \qquad (11)$$

The first term of Eq. 11 is bounded by uniform law of large numbers. To do that, assume that all elements in $\mathbb{H}$ are bounded by constant $C$. First, we consider the case when $f$ is the Pearson divergence. In that case, $BR_f(h)$ is

$$E_{x \sim p(x)}[0.5h(x)^2] - E_{x \sim q(x)}[h(x)]. \qquad (12)$$

We denote the Rademacher complexity of $\mathbb{H}$ as $\mathfrak{R}_{m,q}$ when $x$ is sampled from $q(x)$ and the sample size is $m$. The first term of Eq. 12 is upper bounded by Talagrand's lemma. For any $\delta$ with probability as least $1 - \delta$, we have

$$\sup_{h \in \mathbb{H}} |E_{x \sim p(x)}[0.5h(x)^2] - \frac{1}{m}\sum 0.5h(x_i)^2| \leq C\mathfrak{R}_{m,p}(\mathbb{H}) + 0.5C^2\sqrt{\frac{\log\frac{1}{\delta}}{2m}}, \qquad (13)$$

where samples are taken from $p(x)$ independently. the fact that $0.5h(x)^2$ is C-Lipchitz over the interval $(0, C)$ is used ($C$ is a constant real number). The first term of Eq. 12 is upper-bounded by Talagrand's lemma. For any $\delta$ with probability at least $1 - \delta$, we have

$$\sup_{h \in \mathbb{H}} |E_{x \sim q(x)}[h(x)] - \frac{1}{m}\sum h(x_i)| \leq \mathfrak{R}_{m,q}(\mathbb{H}) + C\sqrt{\frac{\log\frac{1}{\delta}}{2m}}, \qquad (14)$$

where samples are taken from $q(x)$ independently. By combining Eq. 13 and Eq. 14, for any $\delta$ with probability as least $1 - \delta$, the first term of Eq. 11 is bounded by

$$2C\mathfrak{R}_{m,p}(\mathbb{H}) + 2\mathfrak{R}_{m,q}(\mathbb{H}) + (C^2 + 2C)\sqrt{\frac{\log\frac{1}{2\delta}}{2m}}. \qquad (15)$$

When $f$ is the KL-divergence, $BR_f(h)$ is
$$E_{x \sim q(x)}[h(x)] - E_{x \sim p(x)}[\log h(x)]. \qquad (16)$$

The second term of Eq. 16 cannot be bounded because $\log(x)$ is no longer Lipchitz continuous over $(0, C)$. This explains why Pearson divergence is preferable to KL divergence in density ratio estimation. However, if $h(x)$ is lower-bounded by the constant real number $C$, the problem is solved. Using the relative density ratio has the same effect.

In our setting, $r(x)$ is a sigmoid function multiplied by $C$. If $C$ is large, the approximation error, i.e., $BR_f(r_0) - BR_f(r)$ is small. However, there is a possibility that the estimation error increases according to Eq. 15, which may lead to the learning instability.

# E    SUMMARY OF DENSITY RATIO ESTIMATION RESEARCH

In this section, we review researches related to density ratio estimation, entangling them to b-GAN. The Eq. 4 has also been used in class probability density estimation and unnormalized models research. We briefly describe research that is closely connected to density ratio estimation and extract ideas that can also be applied to b-GAN.

## E.1    NOTATION

We summarize notations frequently used in this section.

- $r(x) = p(x)/q(x)$ Density ratios between $p(x)$ and $q(x)$. In b-GAN, $p(x)$ is the probability density function of "real data" and $q(x)$ is the probability density function of "generated data".

- $x_i^{(n)}$ Samples taken from $p(x)$.

- $x_i^{(d)}$ Samples taken from $q(x)$.

- $\mathfrak{D} = (p, q, \pi)$ The joint distribution of $X$ and $Y$. The variable $X$ has the the the mixture distribution of $\pi p + (1 - \pi)q$ and $Y$ is a binary label taking values in $\{-1, +1\}$ which satisfies $\pi = P(Y = 1)$; thus, $(p, q) = (P(X|Y = 1), P(X|Y = -1))$ holds.

- $\eta$ The Bayes optimal estimator $P(Y = 1|X)$ for binary classification.

- $l : \{-1, 1\} \times [0, 1] \to \mathcal{R}$ Loss function

- $h^1 : X \to [0, 1]$ An element of hypothesis sets. In this case, the objective is estimating $P(Y = 1|x)$.

- $h^g : X \to \mathcal{V}$ An element of hypothesis sets. $\mathcal{V}$ is a subset of $\mathcal{R}$. Note that $h^1$ is a special case of $h^g$.

- $\Psi : [0, 1] \to \mathcal{V}$ A link function.

- $k(x, \cdot)$ A positive definite and characteristic (Fukumizu et al., 2004) kernel on the measurable space of $\mathcal{R}$.

- $\mathcal{H}_k$ A reproducing kernel Hilbert space with a kernel $k$ (Steinwart, 2011). An inner product in $H_k$ is $\langle \cdot, \cdot \rangle$.

- $\Phi : \mathcal{R} \to \mathcal{H}_k$ A characteristic function:$x \to k(x, \cdot)$.

- $X_p$ A random variable with probability density function $p$.

- $\mathfrak{B}_f[p\|q]$ Bregman divergence with $f$ between $p$ and $q$.

## E.2    THEORETICAL ASPECTS OF DENSITY RATIO ESTIMATION

Following Kanamori et al. (2012), we expand the explanation of density ratio estimation. As noted in Section 3, the objective function of density ratio estimation is

$$\hat{BR}_f(r_\theta) = \frac{1}{n} \sum_{i=1}^n \left( f'(r_\theta(x_i^{(d)}))r_\theta(x_i^{(d)}) - f(r_\theta(x_i^{(d)})) \right) - \frac{1}{n} \sum_{i=1}^n f'\left(r_\theta(x_i^{(n)})\right), \qquad (17)$$

which is an empirical approximation of Eq. 5. The estimator $\hat{\theta}_n$ is obtained by minimizing Eq. 17. The estimator $\hat{\theta}_n$ is an M-estimator because Eq. 17 is a form of $\frac{1}{n} \sum_{i=1}^n m(x)$, where $m$ is a function of $x$ and $x \sim (x_i^d, x_i^n)$ holds. The estimator $\hat{\theta}_n$ satisfies consistency under mild conditions (see Prop 7.4. (Hayashi, 1997)). The essential condition is the expectation of Eq. 17 is uniquely minimized when $r_\theta$ is equivalent to the true density ratio $r$. We have proved that proposition in Appendix D. Asymptotic normality also holds under suitable conditions (see Prop 7.8. (Hayashi, 1997)).

By differentiating Eq. 17 with respect to $\theta$, the above method can be regarded as a type of moment matching as follows:

$$\frac{\partial BR_f(r_\theta)}{\partial \theta} = \frac{1}{n}\sum_{i=1}^{n} f''(r_\theta(x_i^{(d)})r_\theta'(x_i^{(d)})r_\theta(x_i^{(d)}) - \frac{1}{n}\sum_{i=1}^{n} f''(r_\theta(x_i^{(n)}))r_\theta'(x_i^{(n)}). \quad (18)$$

The estimator $\hat{\theta}_n$ is attained when Eq.18 is equal to 0. By substituting $f''(r_\theta(x_i)r_\theta'(x_i)$ with $s(x_i)$, Eq. 18 is reduced to be

$$\frac{1}{n}\sum_{i=1}^{n} s(x_i^{(d)})r_\theta(x_i^{(d)}) - \frac{1}{n}\sum_{i=1}^{n} s(x_i^{(n)}). \quad (19)$$

The above Eq. 18 is a form of moment matching.[1] What is the optimal $s(x)$? Here, we focus on the variance of the estimator (efficiency). It is known that the Eq. 17 derived from the logistic model is known to be optimal (Qin, 1998). It is a natural consequence because the logistic model can be considered as a maximum likelihood and maximum likelihood reaches the Cramer-Rao bound asymptotically. Typically, general moment matching (GMM) achieves the lower bound when the estimating equation is the score of observations, i.e., when GMM is identical to maximum likelihood.

Efficiency is not the absolute criterion for choosing losses. For example, when there are many outliers in the data, robustness is more important than efficiency. In reality, an M estimator was introduced in the contest of robust statistics (Huber & Ronchetti, 2009).

### E.3  CLASS PROBABILITY ESTIMATION

We have explained density ratio estimation, starting with the Bregman divergence. Importantly, density-ratio estimation is equivalent to class probability estimation. For details, see (Reid & Williamson, 2011; 2010; Dawid & Musio, 2014; Menon & Ong, 2016).

As in Section E.1, we introduce the variable $Y$. The joint distribution of $X$ and $Y$ is denoted as $\mathfrak{D} = (p, q, \pi)$. Class probability estimation can be regraded as a minimization problem of the empirical estimation of the full risk $E_{(X,Y)\sim\mathfrak{D}}[l(Y, h^1)]$, which is denoted $L(h^1; \mathfrak{D}, l)$ ($h^1$ is a hypothesis element). When the hypothesis $h^1$, which minimizes $E_{(X,Y)\sim\mathfrak{D}}[l(Y, h^1)]$, is the Bayes optimal estimator $\eta(x)$ uniquely, such a loss is called a proper loss.

A proper loss is naturally extended to a composite proper loss by introducing a link function $\Phi$. In this case, the objective is estimating $\Phi(\eta(x))$ correctly. The estimator is obtained by the empirical minimization of full risk $E_{(X,Y)\sim D}[l^\Phi(Y, h^g)]$ when $l^\Phi(Y, h^g)$ means $l(Y, \Phi^{-1}(h^g(x)))$ and $\Phi(h^1(x))$ is the same as $h^g(x)$. When the hypothesis $h^g(x)$, which minimizes $E_{(X,Y)\sim D}[l^\Phi(Y, h^g)]$, is the Bayes optimal estimator $\Phi(\eta(x))$ uniquely, such a loss is called a proper composite loss with a link function $\Phi$.

The conditional risk (conditioned on $x$) $E_{Y\sim\eta}[l^\Phi(Y, h^g)]$ can be decomposed to

$$E_{Y\sim\eta}[l^\Phi(Y, h^g)] = \eta L_1(h^g) + (1-\eta)L_{-1}(h^g). \quad (20)$$

The conditional Bayes risk is

$$\eta L_1(\Phi(\eta(x))) + (1-\eta)L_{-1}(\Phi(\eta(x))) = \eta\lambda_{+1}(\eta) + (1-\eta)\lambda_{-1}(\eta), \quad (21)$$

where $\lambda_{+1}(x) = L_1(\Phi(x))$ and $\lambda_{-1}(x) = L_{-1}(\Phi(x))$. We set Eq. 21, i.e., $x\lambda_{+1}(x)+(1-x)\lambda_{-1}(x)$ as $c(x)$. The regret of the composite proper loss can be written using Bregman divergence

$$L(h^g; D, l) - L(\Phi \circ \eta; D, l) \quad = \quad E_X\left[\mathfrak{B}_c\left[\eta(X)\|\Phi^{-1}(h^g(X))\right]\right]. \quad (22)$$

---

[1] In usual moment matching, $s(x)$ is not be dependent on the form of probability density function. However, it depends on the form of $r_\theta$ in this case.

The problem of minimizing the composite proper loss turns out be the density ratio estimation problem by setting $\Phi(x)$ as $u/(1-u)$ and $\pi = 0.5$ (Menon & Ong, 2016). In this case, the LHS of Eq. 22 can be written as $E_{X \sim q} \left[ \mathfrak{B}_{c^\otimes} [\Phi(\eta(X)) \| h^g(X)] \right] (\Phi(\eta(X)) = r(x))$, where $c^\otimes$ is given by

$$c^\otimes : x \to (1+x)c\Big(\frac{x}{1+x}\Big).$$

The equation $E_{X \sim q} \left[ \mathfrak{B}_{c^\otimes} [r(X) \| h^g(X)] \right]$ corresponds to Eq. 4 by substituting $c^\otimes$ with $f$ and $h^g$ with $r_\theta$.

What loss is the optimal loss? The above loss can be written in another form using *weight* by transforming Eq. 22 further. Determining what loss is better has been analyzed from the perspective of *weight*. For example, Reid & Williamson (2010) proposed the "minimal symmetric convex proper loss" for surrogate loss. Regarding density ratio, Menon & Ong (2016) suggest that Pearson divergence is robust because the weight of Pearson divergence is uniform. However, according to their covariate shift experiment, Pearson divergence was not significantly superior to other divergence.

What is the difference between density ratio estimation (class probability estimation) and classification? The objective and assumption differ. As for assumption, in density ratio, the situation where $p(x)$ and $q(x)$ are overlapping would be preferable. However, in classification, the situation where $p(x)$ and $q(x)$ separate would be preferable. In addition, the objective of classification is slightly different from estimating a Bayes rule correctly. Margin loss is widely used in classification rather than zero-one loss. The theoretical guarantee of using margin loss for classification is that it is included in class calibrated loss (Bartlett et al., 2006). However, the margin loss is not equivalent to a proper loss, i.e, it is often not suitable for estimating $\eta$. The condition whereby margin loss is a proper loss is explained by Reid & Williamson (2010). A GAN using margin loss has been proposed (Zhao et al., 2016). Note that using margin loss is not supposed in b-GAN. They succeeded in generating high resolution images.

### E.4 ROBUST LOSS AND DIVERGENCE

What is robust loss and divergence? Basu et al. (1998) proposed a robust divergence called power divergence (Basu et al., 1998), which is given as $\mathfrak{B}_{\nu_\beta}[p \| q]$, where $\nu_\beta$ is

$$\nu_\beta(x) = \frac{1}{\beta(\beta+1)}(x^{\beta+1} - (\beta+1)x + \beta). \tag{23}$$

This is also called $\beta$-divergence (Amari & Cichoki, 2010). The robust estimation equation is derived from power divergence compared to maximum likelihood. By setting $f$ as $\nu_\beta$, robust density ratio estimation has been proposed (Sugiyama et al., 2012). In this case, the objective function is $E_{x \sim q(x)}[\mathfrak{B}_{\nu_\beta}[r \| r_\theta]]$.

Dawid et al. (2015) analyze robust proper loss from the perspective of influence function. They proposed a concept of B-robustness from the perspective of influence function. It is stated that using KL divergence and reversed KL divergence is not robust because the second derivative of $f$ is not bounded at 0. That is a similar conclusion to our analysis in Appendix D.

### E.5 KERNEL METHODS

We assume that $k(x, \cdot)$ is a positive definite kernel. When $X$ is a random variable taking values in $\mathcal{R}$ and $\Psi(X)$ is a random variable taking values in $\mathcal{H}_k$ with a characteristic map $\Psi : x \to k(\cdot, x)$, we can think of the mean of random variable $\Phi(X)$ denoted as $m_X^k$ taking values in $\mathcal{H}_k$, which satisfy $\langle f, m_X^k \rangle = E[\langle f, \Psi(X) \rangle] = E[f(x)](\forall f \in \mathcal{H}_k)$ and $m_X^k(y) = \langle m_X^k, k(\cdot, y) \rangle = E[k(X, y)]$.

If the kernel is characteristic, the bijective from all measures on $\mathcal{R}$ to $H_k$ exists such that a measure $p(x)dx$ corresponds to the mean $m_{X_p}^k$ (Fukumizu et al., 2004). When there are random variables $X_p$ and $X_q$, we can measure the distance between $X_p$ and $X_q$ by calculating $\|m_{X_p}^k - m_{X_q}^k\|_{\mathcal{H}_k}^2$.

As the density ratio estimation methods using kernels, the objective function is the empirical approximation of $\|m_{X_{qr_\theta}}^k - m_{X_p}^k\|_{\mathcal{H}_k}^2$. As generative moment matching networks (GMMN), the objective

function is $\|m_{X_q}^k - m_{X_p}^k\|_{\mathcal{H}_k}^2$ (Dziugaite et al., 2015; Li et al., 2015). GMMNs seem to be superior to b-GAN because they can be trained without density ratio. However, the choice of kernels is difficult. In addition, an autoencoder appears to be required for generating complex data.

### E.6 F-DIVERGENCE ESTIMATION AND TWO SAMPLE TEST

We consider the problem of f-divergence estimation between $p(x)$ and $q(x)$. This is applied straight-forwardly to a two-sample test. Variational f-divergence estimation using Eq. 3 is proposed (Nguyen et al., 2010). In addition, the two step method, i.e., first estimating density ratio and then estimating f-divergence, is proposed (Kanamori et al., 2012). This method is also applied to a two-sample test. The latter method is similar to b-GAN. A kernel two sample test is also introduced calculating $\|m_{X_q}^k - m_{X_p}^k\|_{\mathcal{H}_k}^2$ in (Gretton et al., 2012).

### E.7 UNNORMALIZED MODELS

When the model $p^0(x_m; \phi)$ is unnormalized, the unified method including noise contrastive estimation was proposed in (Pihlaja et al., 2010; Gutmann & Hirayama, 2011; Gutmann & Hyvarinen, 2010).

In this case, the objective is estimating $\theta = \{\phi, C\}$ when the log-likelihood of normalized model $\log p(x; \theta)$ is equal to $\log p^0(x; \phi) + C$ and $C$ is a normalizing constant. Compared to b-GAN, the auxiliary distribution $q(x)$ is known. The parameter $\theta$ can be estimated as a minimization problem of Eq. 17 by replacing $r_\theta(x)$ with $p(x; \theta)/q(x)$ (Pihlaja et al., 2010). As similar algorithm, the method estimating $r_\theta$ first, then estimating $p(x; \theta)$ as $r_\theta(x)q(x)$ is suggested by Gutmann & Hirayama (2011). Note that the latter method is similar to b-GAN.

Pihlaja et al. (2011) analyzed what loss is better by differentiating loss. They experimentally confirmed that noise contrastive estimation is robust with respect to the choice of the auxiliary distribution.

