# Peer review of "b-GAN: Unified Framework of Generative Adversarial Networks"

_ICLR 2017 — rejected_

[Official Review · AnonReviewer1 · rating 4 · confidence 3 · 13 Dec 2016]
**No Title**

This submission introduces a formulation of Generative Adversarial Networks (GANs) under the lens of density ratio estimation, when using Bregman divergences. Even thought GANs already perform density estimation, the motivation of using Bregman divergences is to obtain an objective function with stronger gradients. I have three concerns with this submission.

First, the exposition of the paper must be significantly improved. The current version of the manuscript is at some points unreadable, and does a poor job at motivating, describing, and justifying the contributions.

Second, the authors scatter a variety of alternatives and heuristics throughout the description of the proposed b-GAN. This introduces a great amount of complexity when it comes to understanding, implementing, and using b-GAN. Further work is necessary to rule out (in a principled manner!) many of the proposed variants of the algorithm.

Third, it is next to impossible to interpret the experimental results, in particular Figures 2, 3, 4. The authors claim that these figures show that "learning does not stop", but such behavior can also be attributed to the typical chaotic dynamics of GANs. Even after reading Appendix A, I am left unconvinced on whether the proposed approach provides with any practical advantage (even no comparison is offered to other GAN approaches with similar architectures).

Overall, I believe this submission calls for significant improvements before being considered for publication.

[Official Review · AnonReviewer3 · rating 6 · confidence 4 · 16 Dec 2016]
**Interesting paper on connections between GANs and density ratio estimation research**

This paper proposes b-GAN, which trains a discriminator by estimating density ratio that minimizes Bregman divergence. The authors also discuss how b-GANs relate to f-GAN and the original GAN work, providing a unifying view through the lens of density ratio estimation. 

Note that the unifying view applies only to GAN variants which optimize density ratios. In general, GANs which use MMD in the discriminator step do not fit in the b-GAN framework except for special choices of the kernel. 

I was a bit confused about the dual relationship between f-GAN and b-GAN. Are the conditions on the function f the same in both cases? If so, what's the difference between f-GAN and b-GAN (other than the fact that the former has been derived using f-divergence and the latter has been derived using Bregman divergence)?

One of the original claims was that b-GANs optimize f-divergence directly as opposed to f-GAN and GAN. However, in practice, the authors optimize an approximation to the f-divergence; the quality of the approximation is not quantified anywhere, so b-GAN doesn't seem more principled than f-GAN and GAN.

The experiments left me a bit confused and were not very illuminating on the choice of f. 

Overall, I liked the connections to the density ratio estimation literature. The appendix seems like a scattered collection right now. Some re-writing of the text would significantly improve this paper.

[Official Review · AnonReviewer4 · rating 5 · confidence 3 · 19 Dec 2016]

In this paper, the authors extend the f-GAN by using Bregman divergences for density ratio matching. The argument against f-GAN (which is a generalization of the regular GAN) is that the actual objective optimized by the generator during training is different from the theoretically motivated objective due to gradient issues with the theoretically motivated objective. In b-GANS, the discriminator is a density ratio estimator (r(x) = p(x) / q(x)), and the generator tries to minimize the f-divergence between p and q by writing p(x) = r(x)q(x).

My main problem with this paper is that it is unclear why any of this is useful. The connection to density estimation is interesting, but any derived conclusions between the two seem questionable. For example, in previous density estimation literature, the Pearson divergence is more stable. The authors claim that the same holds for GANS and try to show this in their experiments. Unfortunately, the experiments section is very confusing with unilluminating figures. Looking at the graph of density ratios is not particularly illuminating. They claim that for the Pearson divergence and modified KL-divergence, "the learning did not stop" by looking at the graph of density ratios. This is completely hand-wavey and no further evidence is given to back this claim. Also, why was the normal GAN objective not tried in light of this analysis? Furthermore, it seems that despite criticizing normal GANs for using a heuristic objective for the generator, multiple heuristics objectives and tricks are used to make b-GAN work.

I think this paper would be much improved if it was rewritten in a clear fashion. As it stands, it is difficult to understand the motivation or intuition behind this work.

[Final Decision · Program Chairs · 06 Feb 2017]
**ICLR committee final decision**

The paper may have an interesting contribution, but at present its motivation, and the presentation in general, are not clear enough. After a re-write, the paper might become quite interesting and should be submitted to some other forum.